# Factors Related to Stunting Incidence in Toddlers with Working Mothers in Indonesia

**DOI:** 10.3390/ijerph191710654

**Published:** 2022-08-26

**Authors:** Agung Dwi Laksono, Noor Edi Widya Sukoco, Tety Rachmawati, Ratna Dwi Wulandari

**Affiliations:** 1National Research and Innovation Agency, Government of Indonesia, Jakarta 10340, Indonesia; 2Faculty of Public Health, Universitas Airlangga, Surabaya 60115, Indonesia

**Keywords:** community nutrition, working mother, nutritional status, stunted toddler, public health

## Abstract

Previous studies have suggested that a toddler stunting is closely related to maternal characteristics. Working mothers, as a group, are vulnerable to having a stunted toddler. The present research aimed to analyze factors related to stunting incidence in toddlers with working mothers in Indonesia. The study sampled 44,071 toddlers with working mothers. The final stage used a multinomial logistic regression test. The study found that working mothers living in rural areas have a higher probability of having stunted or severely stunted toddlers. Maternal age partially affects the incidence of stunted toddlers in Indonesia. Mothers in the ≤19 age group are 1.461 (95% CI 1.140–1.872) times more likely than those in the ≥45 group to have a severely stunted toddler. Those who were never married were 1.433 (95% CI 1.006–2.043) times more likely than those who were divorced/widowed to have a severely stunted toddler. A married mother was 0.734 (95% CI 0.617–0.872) times less likely to have a severely stunted toddler than a divorced/widowed mother. Better education is protective against working mothers having stunted toddlers. Moreover, the present study found that the toddler’s age determined the incidence of stunted toddlers. This study concluded that there are five variables related to stunting incidence in toddlers with working mothers in Indonesia: residence, age, marital status, education, and toddler age.

## 1. Introduction

According to the World Health Organization (WHO), children are defined as stunted if their height by their age is more than two standard deviations below the WHO Child Growth Standards median. The z-score for height by age under minus two standard deviations from the global growth reference is noted in [1]. During childhood, stunting is the best indicator available to measure toddlers’ well-being. This condition accurately reflects the environmental context and social inequality [2].

Stunted toddlers experience chronic nutritional problems caused by multiple factors. Socio-economic conditions, maternal nutrition during pregnancy, infant illness, and lack of nutritional intake in infants can become determinants of toddler stunting, leading to difficulties in achieving optimal physical and cognitive development [1]. Insufficient food availability to meet nutritional intake needs and the emergence of infectious diseases are the most direct and frequent causes of growth failure in children under five [3].

The impact of stunting on children under five also has consequences for health status in the future. A previous study found that stunting in early life would have detrimental functional implications, including poor cognition and lost productivity. Furthermore, if weight gain in childhood accompanies stunting, it increases the risk of chronic disease associated with excess nutritional status [4].

Indonesia has recorded that the prevalence rate of stunted toddlers is relatively high. The information recorded in the Indonesia Basic Health Survey in 2007, 2013, and 2018, showed that the national prevalence of stunting for children under five was 36.8%, 37.2%, and 30.8%, respectively [5]. Although this figure has fluctuated and tends to fall, the prevalence is still above 30%. The records were the highest among regional countries in Southeast Asia [5,6]. Global Health Observatory records indicate that, globally, around 21.9%, or nearly 150 million children under five, are stunted. We expected that the prevalence of stunting could fall to 17.5% by 2030 to reduce the adverse effects of stunting [7]. In general, most stunting in Indonesia is still high according to the limits set by WHO, which is >20%. The Indonesian government aims to reduce stunting to 14% by 2024 by following the National Medium Term Plan [8].

Stunting can start during pregnancy, and the role of a mother also determines the nutritional status of children under five children after birth. Previous studies have provided insight into the relationships between a mother’s health, nutrition, and sociodemography and the incidence of stunting in children under five [9,10,11]. The cultural context of Asian countries, including Indonesia, dictates that domestic responsibility and child care rely on mothers [12,13,14].

In some low-income families, mothers are also forced to share the responsibility of earning a living by working apart from being responsible for domestic affairs. Consequently, the availability of time and attention for childcare is reduced. Thus, the condition of mothers of children under five also being workers can become a particular problem for children’s growth and nutritional status [15]. Based on the background description, this study aims to analyze factors related to stunting incidence in toddlers with working mothers in Indonesia.

## 2. Materials and Methods

### 2.1. Study Design

The cross-sectional study uses secondary data from the 2017 Indonesia Nutritional Status Monitoring survey. The survey was deployed on a national scale using the multi-stage cluster random sampling method. The Nutrition Directorate of the Indonesian Ministry of Health conducted the survey [16]. The study population was all working mothers with children under five in Indonesia. As many as 44,071 working mothers were involved in this study.

### 2.2. Dependent Variable

The study used toddlers’ nutritional status (stunting) as a dependent variable. Stunting is an indicator of nutritional status that is assessed based on the height of a child that has been reached at a certain age. According to WHO growth standards, the height for age indicator is determined based on the z-score, or the height deviation from average height. The limit for the nutritional status category for children under five, according to the height for age index provided by the WHO is [16]:-Severely stunted: <−3.0 SD-Stunted: −3.0 SD to −2.0 SD-Normal: ≥−2 SD.

### 2.3. Independent Variables

The present study employed five independent variables, i.e., the type of residence, maternal age, maternal marital status, maternal education level, and the toddler’s age. The kind of residence consists of two types: urban and rural. The study refers to the Central Statistics Agency for categorizing the type of residence.

The study determines maternal age based on the respondent’s last birthday. Maternal age consists of seven groups: ≤19, 20–24, 25–29, 30–34, 35–39, 40–44, and ≥45 years of age. There are three maternal marital status groups: never married, married, and widowed/divorced. Meanwhile, the study calculates maternal education based on the most recent certificate possessed by the moms of children under five. Maternal education consists of four levels: primary school and under, junior high school, senior high school, and college.

### 2.4. Data Analysis

The present study performed bivariate analyses using chi-square analyses to test dichotomous variables. Meanwhile, the study used *t*-tests for continuous variables. In the final stage, the study conducted a multivariate analysis using multinomial logistic regression to determine the variables affecting the incidence of stunted toddlers with working mothers in Indonesia. The author used the IBM SPSS 26 program for the entire statistical analysis process. Moreover, the study employed ArcGIS 10.3 (ESRI Inc., Redlands, CA, USA) to map stunted toddlers among working mothers in the province of Indonesia in 2017. The Indonesian Bureau of Statistics provided a shapefile of administrative border polygons for the job.

### 2.5. Ethical Approval

The 2017 Indonesia Nutritional Status Monitoring survey has an ethics license approved by the national ethics committee from the National Institute of Health Research and Development (ethics number: LB.02.01/2/KE.244/2017). The survey used informed consent during data collection, which took into account aspects of the procedure for data collection, the voluntary nature of the survey, and confidentiality.

## 3. Results

The analysis results found that the average prevalence of stunted toddlers (stunted and severely stunted) with working mothers in Indonesia was 30.9%. Meanwhile, Figure 1 shows a map of the distribution of stunted toddlers among working mothers. The map shows that the distribution of stunted toddlers does not show a particular pattern based on region. Only the Java-Bali region, as the center of government, offers a lower prevalence of stunted toddlers.

Table 1 provides statistical descriptions of the characteristics of the nutritional status of toddlers with working mothers in Indonesia. Table 1 shows that working mothers predominantly live in rural areas. Based on age groups, mothers’ age falling within the 25–29-year-old age group was associated with the distinct and stunted categories; meanwhile, the 30–34 age group occupied the severely stunted type.

Based on maternal marital status, married mothers dominated all categories of the nutritional status of toddlers. Meanwhile, based on maternal education level, mothers with senior high school education levels led various nutritional status categories for toddlers. Finally, based on the toddler’s age, on average, toddlers with stunted and severely stunted nutritional statuses were older than toddlers with normal nutritional status.

Table 2 presents the multinomial logistic regression test results from analyses aimed at determining the variables affecting the incidence of stunted toddlers among working mothers in Indonesia. This multinomial logistic regression test uses the “normal (≥−2 SD)” nutritional status of children under five as a reference group.

These analyses show that working mothers living in rural areas had a higher probability of having stunted or severely stunted toddlers. Working mothers living in urban areas were 0.762 less likely than those living in rural areas to have a stunted toddler (AOR 0.762; 95% CI 0.716–0.811). Working mothers who lived in urban areas were 0.613 times less likely than those who lived in rural areas to have severely stunted toddlers (AOR 0.613; 95% CI 0.563–0.667).

Table 2 shows that the age group partially affected the incidence of stunted toddlers with working mothers in Indonesia. Working mothers in the ≤19 age group were 1.461 times more likely than those in the ≥45 age group to have a severely stunted toddler (AOR 1.461; 95% CI 1.140–1.872).

Based on marital status, the results found that a never-married mother was 1.433 times more likely than a divorced/widowed mother to have a severely stunted toddler (AOR 1.433; 95% CI 1.006–2.043). On the other hand, a married mother was 0.734 times less likely than a divorced/widowed mother to have a severely stunted toddler (AOR 0.734; 95% CI 0.617–0.872). These analyses showed that marriage was a protective factor against working mothers having stunted toddlers in Indonesia.

Based on the education level, Table 2 shows that working mothers with primary school education were 1.692 times more likely than those with a college education to have a stunted toddler (AOR 1.692; 95% CI 1.571–1.822). Working mothers with primary school education were 2.435 times more likely than those with a college education to have severely stunted toddlers (AOR 2.435; 95% CI 2.216–2.676).

Meanwhile, working mothers with a junior high school education were 1.546 times more likely than those with a college education to have a stunted toddler (AOR 1.546; 95% CI 1.427–1.674). Working mothers with junior high school education were 1.727 times more likely than those with a college education to have severely stunted toddlers (AOR 1.727; 95% CI 1.555–1.917).

Finally, working mothers with senior high school education were 1.313 times more likely than those with a college education to have a stunted toddler (AOR 1.313; 95% CI 1.222–1.411). Working mothers with senior high school education were 1.416 times more likely than those with a college education to have severely stunted toddlers (AOR 1.416; 95% CI 1.022–1.026).

These analyses show that a better level of education was a protective factor against working mothers having stunted toddlers. Moreover, the study also found that the toddler’s age determined the incidence of stunted toddlers among working mothers in Indonesia.

## 4. Discussion

The analysis found that working mothers living in rural areas had a higher probability of having stunted or severely stunted toddlers. The development disparity between urban and rural areas in Indonesia has continued for a long time. Growth in urban areas looks more advanced than in rural areas. The government provides public facilities, but more dense urban areas make private parties compete to build facilities in urban areas [17,18]. The gap occurs in general development and includes access to health services [19,20]. The results showed that working mothers’ age affected the incidence of stunted toddlers in Indonesia. The results reinforce findings made in previous studies in Rwanda and India [21,22]. Meanwhile, a study in India specifically explained that the mother’s age when she first married and the mother’s age at delivery were determinants of stunting; the younger the maternal age at the time of marriage, the riskier it was to have severely stunted children [21].

The present study found that married mothers had the highest possibility of not having stunted toddlers in Indonesia. In the context of the social system in Indonesia, men are responsible for earning a living, while women are responsible for domestic affairs. With a good level of education, they also have a more significant opportunity to acquire a better/decent job, which impacts the social economy [14,23]. Unmarried working mothers held two responsibilities, which reduced their attention on the children [6,15,24]. On the other hand, divorce creates chaos for livelihoods and disrupts food availability in the household [25].

The results showed that a better education level was a protective factor against working mothers having a stunted toddler. Working mothers’ higher education level is closely related to their capacity to care for children [26,27]. The higher the mother’s education level, the better the child’s growth. Working mothers with high education levels automatically have better knowledge regarding meeting the needs of children [28,29]. This condition applies physically, mentally, and socially because the caregiver’s mental health and parents’ parenting practices are influential [30]. These findings are in line with another study conducted in Indonesia [31].

On the other hand, research evaluating the same study topic in India, Ethiopia, Tanzania, Uganda, and Vietnam has also found similar results; the mother’s level of education is a determinant of stunting in children [32,33]. The positive effect of education level applies to children’s nutritional status and spreads to other health sectors [34,35,36]. Meanwhile, several studies have identified poor education as a barrier to health program output, and poor education can affect access to knowledge, especially in regard to health and nutrition [37,38,39,40].

Apart from the mother’s age, the analysis results also found that the toddler’s age was a determinant of the nutritional status of toddlers with working mothers in Indonesia. Other studies in Indonesia also noted consistent research results being reported at the national level [25]. Moreover, studies in several countries with the same theme also indicate similar results, including Uganda and India [41,42].

This study proves that toddler stunting is the result of multi-dimensional factors that are closely related to maternal characteristics. Based on the present study’s findings, strengthening the education sector and family resilience are critical interventions for reducing the prevalence of stunting [25].

Furthermore, a study using secondary data from the Demographic and Health Survey (DHS) in Peru found no significant relationship between maternal employment status and stunting in children aged 6 to 36 months. However, in a multivariate analysis, the study found that stunting children from mothers who performed unpaid work had a higher risk of stunting. The results indicated a significantly higher prevalence of stunting in children whose mothers completed unpaid work (12.4%) (OR 1.38; 95% CI 1.2–1.6; *p* < 0.001) compared with paid working mothers. These findings can provide support for implementing education programs and labor policies to reduce the prevalence of stunting among children [43].

### Study Limitation

Due to the limitations in the data from the 2017 Indonesia Nutritional Status Monitoring survey, this study did not involve previously known variables as determinants of toddler stunting. Among them are the characteristics of the mother, in the form of height, body mass index (BMI), antenatal visits [44], and a history of smoking and alcohol consumption [45]; as well as the characteristics of the toddler in the form of gender, weight at birth [46], history of childhood illness, and the breastfeeding process [47].

## 5. Conclusions

The present study concluded that all of the tested variableswere related to the incidence of stunted toddlers with working mothers in Indonesia. The type of residence, maternal education level, and toddler’s age affected toddler stunting among children under age five with working mothers in Indonesia. Meanwhile, the residence, maternal age group, maternal marital status, maternal education level, and toddler’s age were related to severely stunted toddlers with working mothers in Indonesia.

## Figures and Tables

**Figure 1 ijerph-19-10654-f001:**
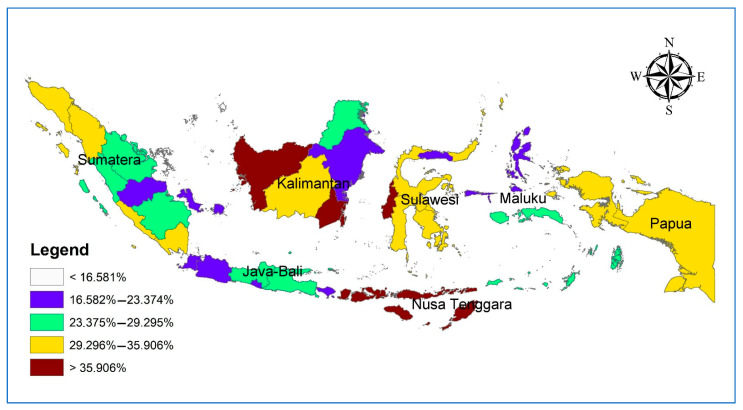
Distribution map of stunted toddlers with working mothers in Indonesia (*n* = 44,071).

**Table 1 ijerph-19-10654-t001:** Descriptive statistics for stunted toddlers with working mothers in Indonesia, 2017 (*n* = 44,071).

Variables	Nutritional Status of Toddlers						*p*-Value
	Normal		Stunted		Severely Stunted		
	*n*	%	*n*	%	*n*	%	
Type of residence							<0.001
• Urban	7455	24.5%	1609	18.8%	745	14.7%	
• Rural	22,998	75.5%	6941	81.2%	4323	85.3%	
Maternal age							<0.001
• ≤19	693	2.3%	168	2.0%	157	3.1%	
• 20–24	4212	13.8%	1174	13.7%	747	14.7%	<0.001
• 25–29	8782	28.8%	2411	28.2%	1345	26.5%	
• 30–34	8556	28.1%	2366	27.7%	1399	27.6%	
• 35–39	5342	17.5%	1520	17.8%	881	17.4%	
• 40–44	2037	6.7%	665	7.8%	383	7.6%	
• ≥45	831	2.7%	246	2.9%	156	3.1%	
Maternal marital status							<0.001
• Never married	170	0.6%	65	0.8%	55	1.1%	
• Married	29,606	97.2%	8241	96.4%	4841	95.5%	
• Divorced/Widowed	677	2.2%	244	2.9%	172	3.4%	
Maternal education level							<0.001
• Primary school and under	7734	25.4%	2762	32.3%	2025	40.0%	
• Junior high school	5437	17.9%	1738	20.3%	981	19.4%	
• Senior high school	9644	31.7%	2546	29.8%	1339	26.4%	
• College	7638	25.1%	1504	17.6%	723	14.3%	
Toddler’s age (in months; mean)	30,453	(24.83)	8550	(31.77)	5068	(30.90)	<0.001

**Table 2 ijerph-19-10654-t002:** The results of multinomial logistic regression analyses of stunted toddlers with working mothers in Indonesia in 2017 (*n* = 44,071).

Predictors	Stunted			Severe Stunted		
	AOR	95% CI		AOR	95% CI	
		Lower Bound	Upper Bound		Lower Bound	Upper Bound
Type of place: Urban	*** 0.762	0.716	0.811	*** 0.613	0.563	0.667
Type of place: Rural	-	-	-	-	-	-
Age: ≤19	1.035	0.826	1.295	** 1.461	1.140	1.872
Age: 20–24	1.119	0.954	1.314	1.193	0.985	1.446
Age: 25–29	1.141	0.979	1.330	1.121	0.932	1.348
Age: 30–34	1.103	0.947	1.286	1.155	0.960	1.388
Age: 35–39	1.077	0.921	1.259	1.076	0.891	1.299
Age: 40–44	1.141	0.962	1.352	1.095	0.891	1.346
Age: ≥45	-	-	-	-	-	-
Marital: Never married	1.242	0.896	1.723	* 1.433	1.006	2.043
Marital: Married	0.877	0.754	1.021	*** 0.734	0.617	0.872
Marital: Divorced/Widowed	-	-	-	-	-	-
Education: Primary school	*** 1.692	1.571	1.822	*** 2.435	2.216	2.676
Education: Junior high school	*** 1.546	1.427	1.674	*** 1.727	1.555	1.917
Education: Senior high school	*** 1.313	1.222	1.411	*** 1.416	1.285	1.560
Education: College	-	-	-	-	-	-
Toddler’s age (in months)	*** 1.027	1.025	1.028	*** 1.024	1.022	1.026

Confidence Interval (CI) of 95% for adjusted odds ratio (AOR); * *p <* 0.05; ** *p* < 0.01; *** *p* < 0.001.

## Data Availability

The author cannot publicly share the data because the Ministry of Health of the Republic of Indonesia, which owns the information, does not permit it. The requested data set is available from the contact https://www.litbang.kemkes.go.id/layanan-permintaan-data-riset/ (accessed on 21 July 2022) for researchers who meet the criteria for access to confidential data.

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
