# Peer review of "Factors Related to Stunting Incidence in Toddlers with Working Mothers in Indonesia"

_ijerph, 2022, doi:10.3390/ijerph191710654_

Round 1
Reviewer 1 Report
Dear authors,
you can find my corrections and suggestion in the attached file.
My best regards.

Author Response
Dear authors,
I revised the manuscript Factors Related to Stunting Incidence in Toddlers with Working Mothers in Indonesia.
This article is a bit chaotic. An extensive English language editing is required and a deepen discussion must be given.
Results are presented not so clearly. In the next lines you can find punctual corrections and suggestions to improve the quality of your work.
ABSTRACT
Line 20: “Moreover, the study found that the toddler's age determined the incidence of stunted toddlers”. I don’t understand, please consider to better express.
Please explicit the age of toddlers. It is too generic to speak about toddlers.
INTRODUCTION
Line 26: please explicit the acronym WHO. I know everyone knows for what it stands for, anyway is better to explicit. Moreover, the definition you provide here is very generic and imprecise. I suggest you to reformulate the sentence. Even from a grammatical point of view something doesn’t work properly. Please revise also considering the WHO definition for stunting in childhood. “(…) Children are defined as stunted if their height-for-age is more than two standard deviations below the WHO Child Growth Standards median.” https://www.who.int/news/item/19-11-2015-stunting-in-a-nutshell
Response: the sentence was revised as suggested.
Line 32: please revise the sentence form. Hard to read.
Response: the sentence was revised as suggested.
Line 40: What do you mean for functional implications? I don’t understand how low educational levels are linked to the detrimental effects of stunting. I think this issue has implications with the causes of stunting. Please explain better
Response: the sentence was revised as suggested.
Line 45: I would eliminate in Indonesia at the end of the sentence
Response: the sentence was revised as suggested.
Line 50: why do you expect this fall? Please detail more on this
Response: … to reduce the adverse effects of stunting.
Line 51-52: please revise the grammar. A verb is missing in my opinion.
Response: the sentence was revised as suggested.
Line 54: mother’s characteristics you cite are those reported in line 55? If not, you must report some of them. This passage is confusing. In fact, from line 55, it seems that mother's health, nutrition, and sociodemographic are causing stunting among mothers. I think it would be better if you dedicate some sentences for giving more details about the spreading of stunting. Here you must explain that stunting can begin during the pregnancy. In this way mother’s role in its development would be clearer. After this you can pass to the working mother’s condition; how this affects stunting, with other considerations.
Response: the sentence was revised as suggested.
Line 57: please revise the language use. Please use the active form and prefer more direct expressions. E.g. The cultural context of Asian countries, including Indonesia, foresees that domestic responsibility and child care rely on mothers.
Response: the sentence was revised as suggested.
Line 58: add child after five. Please revise all the sentence. It is very hard to read. Simplify the form eliminating useless information; for example, it is obvious that working mothers are female workers… Also, put in evidence the consequence. E.g. Working mothers have reduced time for parenting care, so they cannot actively control the nutritional status.
I think, in any case you must give some information more about the socio-economic state of Indonesian stunted child. I mean, why mothers start working if, as you stated, is socially accepted that they take care of child and more in general of the family exigencies. I try to explain me better. It would be nice if you can detail about the reason why mothers start working. There is an economic urgency among Indonesian families? Or the fact that mothers start working depends on social and cultural changes?
Response: the sentence was revised as suggested.
Materials and Methods
Lines 65-71: please consider rewriting this paragraph. Verbal tenses are wrongly used. The sentences are too redundant and confuse.
Response: the sentence was revised as suggested.
Line 74: “(…) is assessed based on height for age or the height of a child who is reached at a certain age” It seems a repletion. Consider eliminating one of the two definitions. Also, please carefully check the writing.
Response: the sentence was revised as suggested.
Line 85: a reference is needed
Response: the reference is already in that part.
Line 98: version and year of the software you used
Response: The author used the IBM SPSS 26 program for the entire statistical analysis process.
Line 104: you must detail more on the ethical committee approval. Which is the name of the approving institution?
Response: The 2017 Indonesia Nutritional Status Monitoring has an ethics license approved by the national ethics committee from the National Institute of Health Research and Development.
Results
Why did you combine stunted+severe stunted? You must explicit, giving reasons for this union.
Response: The author combined the stunted to simplify the figure.
Figure 1: I suggest you to put in evidence names of the regions. From the colors is not so easy to get where Java-Bali is for example.
Response: the figure was revised as suggested.
TABLE 1: revise the format.
Line 135: you should give some information about the “normal” nutritional status.
I don’t understand how the variable toddler’s age influences the probability of a toddler to be stunted or severe stunted. Also, how this is connected to mothers’ characteristics. You should give more details on toddlers’ age and correlate this characteristic to those of mothers. Putting the mean value in months does not give a real information.
Response: The age of under five is related to the period of care by the mother.
DISCUSSION
Line 174: something missing in this sentence. Please reformulate.
Response: the sentence was revised.
Line 183: is this finding still linked to the previous study cited? If not, how did you get this information?
Response: yes, it is.
Line 185: married?
Response: The study found married mother has the highest possibility not to have stunted toddlers in Indonesia.
Line 190: in which sense it reduces children’ attention? Are you referring to cognitive ability of children or mothers’ attention to their exigencies? Please rephrase
Response: the sentence was revised.
Line 192: the article a is missing
Response: the sentence was revised.
Line 214: what is DHS? You must specify
Response: the sentence was revised as suggested.

Reviewer 2 Report
ijerph-1853020_ Factors Related to Stunting Incidence in Toddlers with Working Mothers in Indonesia
Thank you very much in order to allow me to review the article entitled “Factors Related to Stunting Incidence in Toddlers with Working Mothers in Indonesia” (ijerph-1853020).
The title reflects the content of the manuscript.
The aims of this manuscript is to analyze factors related to stunting incidence in toddlers with working mothers in Indonesia.
This study is conducted in a country with a high prevalence of stunting in toddler as the information recorded in the Indonesia Basic Health Survey in 2007, 2013, and 2018, shows that the national prevalence of stunting for children under five in Indonesia is 36.8% 37.2 %, and 30.8%, respectively. Focusing on the children of working women.
The introduction is clear and well thought out.
The methodology is a cross-sectional design carried out in 2017. This is a limitation of the study since we are in 2022 and the country in which it is carried out presents a fairly accelerated development. However, it should also be noted that there may not be more up-to-date data on this subject. This is a question to the authors.
The sample size is 44,071 working women but the authors do not indicate to which labor sector they belong; it would be convenient to incorporate it.
The methodology is well described, although it would be interesting to know if they have more children since the first pregnancy is not the same as when the woman works she is pregnant and she has to take care of other children.
The results are clearly presented in very informative tables. In table two at the bottom of the table, the adjustment variable used should be indicated.
The discussion should take into account the correlation between a primary education and a more precarious job, which usually involves greater physical effort.
This study reinforces the importance of care during pregnancy.
Author Response
Thank you very much in order to allow me to review the article entitled “Factors Related to Stunting Incidence in Toddlers with Working Mothers in Indonesia” (ijerph-1853020).
The title reflects the content of the manuscript.
The aims of this manuscript is to analyze factors related to stunting incidence in toddlers with working mothers in Indonesia.
This study is conducted in a country with a high prevalence of stunting in toddler as the information recorded in the Indonesia Basic Health Survey in 2007, 2013, and 2018, shows that the national prevalence of stunting for children under five in Indonesia is 36.8% 37.2 %, and 30.8%, respectively. Focusing on the children of working women.
The introduction is clear and well thought out.
The methodology is a cross-sectional design carried out in 2017. This is a limitation of the study since we are in 2022 and the country in which it is carried out presents a fairly accelerated development. However, it should also be noted that there may not be more up-to-date data on this subject. This is a question to the authors.
Response: Even though it uses 2017 data, this data set is the latest raw data allowed to be publicly disclosed by the ministry of health.
The sample size is 44,071 working women but the authors do not indicate to which labor sector they belong; it would be convenient to incorporate it.
Response: The study sample is a mother who claims to work, no matter what type of work.
The methodology is well described, although it would be interesting to know if they have more children since the first pregnancy is not the same as when the woman works she is pregnant and she has to take care of other children.
Response: Due to the limited data from the 2017 Indonesia Nutritional Status Monitoring, this study did not involve previously known variables as determinants of stunted toddlers.
The results are clearly presented in very informative tables. In table two at the bottom of the table, the adjustment variable used should be indicated.
Response: Since the analysis uses a multivariate approach, all independent variables adjust to each other.
The discussion should take into account the correlation between a primary education and a more precarious job, which usually involves greater physical effort.
Response: Thank you for your suggestion. Due to the limited data from the 2017 Indonesia Nutritional Status Monitoring, this study did not involve previously known variables as determinants of stunted toddlers.
This study reinforces the importance of care during pregnancy.

Round 2
Reviewer 1 Report
Dear authors,
thank you for following the indications. In my opinion the current version is better and easier to read.
I think that in general, the manuscript does not really give back a great piece of new information. Moreover your data are of a quite old database and in many passages you also declare that.
At any rate, some nice findings are present and may justify a continuation of your research.
Even if the global quality must be increased, I think that your work can be considered for pubblication.
My best regards